# The Modification of Offspring Stress-Related Behavior and the Expression of *Drd1, Drd2*, and *Nr3c1* by a Western-Pattern Diet in *Mus Musculus*

**DOI:** 10.3390/ijms23169245

**Published:** 2022-08-17

**Authors:** Nikki Clauss, Kelsey Brass Allen, Katie D. Billings, Mikayla D. M. Tolliver, Ray Garza, Jennifer Byrd-Craven, Polly Campbell

**Affiliations:** 1Department of Cellular and Integrative Physiology, University of Texas Health Science Center at San Antonio, 7703 Floyd Curl Dr., San Antonio, TX 78229, USA; 2College of Medicine, University of Oklahoma Health Science Center, 1100 N Lindsay Ave, Oklahoma City, OK 73104, USA; 3Department of Science and Mathematics, Mississippi College, Box 4053, Clinton, MS 39058, USA; 4Department of Psychological Science, 216 Memorial Hall, University of Arkansas, Fayetteville, AR 72701, USA; 5Department of Psychology and Communication, Texas A&M International University, 5201 University Blvd, Laredo, TX 78041, USA; 6Department of Psychology, Oklahoma State University, 116 Psychology Building, Stillwater, OK 74078, USA; 7Department of Evolution, Ecology, and Organismal Biology, University of California, Riverside, 900 University Ave, Riverside, CA 92521, USA

**Keywords:** diet, development, stress, maternal behavior, sex differences, gene expression

## Abstract

The impact of early developmental experience on neurobiological pathways that may contribute to the association between diet and behavior have not yet been elucidated. The focus of the current study was to determine whether the impact of prenatal stress (PS) could be mitigated by a diet that stimulates the same neuroendocrine systems influenced by early stress, using a mouse model. Behavioral and genetic approaches were used to assess how a Western-pattern diet (WPD) interacts with PS and sex to impact the expression of anxiety-like behavior in an open-field arena, as well as the expression of the glucocorticoid receptor in the hippocampus, D1 dopamine receptors in the nucleus accumbens, and D2 dopamine receptors in the ventral tegmental area. Overall, the results demonstrated that a prenatal WPD mitigates the effects of maternal stress in dams and offspring. These results help to elucidate the relationship between pre- and post-natal nutrition, gene expression, and behaviors that lead to long-term health effects.

## 1. Introduction

Critical periods in an organism’s development may play a particularly important role in the establishment of obesity-related eating behaviors and anxiety-like behavioral phenotypes. During such critical periods, the biological settings of a developing system are especially responsive to social and biological cues and are therefore open to modification in response to environmental experience [1]. Importantly, many critical periods overlap with periods of placental and lactational maternal provisioning, which may indicate that these periods are windows of opportunity for the transfer of phenotypic information from one generation to the next. Maternal stress impacts developing offspring during these critical periods via several mechanisms, including hormones, nutrients, and altered maternal care.

Maternal antenatal stress and postpartum behavior are significant sources of early environmental adversity. Environmental adversity occurring early in development is a crucial factor contributing to disease susceptibility and anxiety states in later life [2]. Antenatal stressors in early human development are associated with alterations in cognitive, behavioral, and emotional processes [3,4]. In rodents, stressors induce anxiety- and depression-like behaviors in pregnant dams, which lead to a reduction in attentive maternal behaviors [5,6,7]. Importantly, the quality and quantity of care that mothers provide can have lasting effects on offspring physiology, behavior, and neural gene expression [7].

Additionally, the exposure of pregnant dams to stress results in dramatic increases in corticosterone, which has a profound impact on offspring brain development. Stressors engage a network of limbic structures that are responsive to both interoceptive and exteroceptive inputs, including the hippocampus [8]. The recruitment of this stress network depends on the release and action of corticotropin releasing factor (CRF) from the paraventricular nucleus of the hypothalamus, and subsequent release of glucocorticoids from the adrenal cortex. Acute and chronic stressors both lead to increases in synaptic activity and dendritic bushing in the amygdala and anterior cingulate cortex (areas of the brain involved in the processing of emotional stimuli) and reduce synaptic contacts in the hippocampus and prefrontal cortex (areas of the brain involved in stimulus interpretation, planning, and behavioral control) [9,10]. Stress exposure during late pregnancy is associated with permanent modifications of hypothalamic pituitary adrenal axis (HPAa) functioning in the offspring, including prolonged corticosterone responses to stress [11], and a predisposition for limbic-biased stress responses. Evidence suggests that the pattern of HPAa activity exhibited by prenatally stressed offspring is associated with an increased likelihood of reward-based energy consumption [12].

Chronic stress and reduced maternal care are associated with decreased extracellular dopamine in the nucleus accumbens shell and medial prefrontal cortex, leading to altered sensorimotor gating and increased anxiety-like behavior [13,14]. This stress-related reduction in extracellular dopamine may be mitigated by exogenous substances, such as highly palatable food. Eating highly palatable foods after a stressor attenuates the stress response such that the expected stress induced CRF expression and secretion of adrenocorticotropic hormone (ACTH) and glucocorticoids are reduced [15,16,17]. The apparent stress-reducing properties of highly palatable food appear to be associated with increases in dopamine release, as dopamine stimulation leads to downstream increases in the expression of glucocorticoid receptor mRNA and enhances negative feedback regulation of the hypothalamic stress response [7]. Indeed, people who eat more comfort food have dampened HPAa stress responses [18], and rats that are trained to earn vanilla sugar pellets and then exposed to stress have basal levels of dopamine accumulation in the nucleus accumbens shell comparable to their counterparts who received sugar pellets without subsequent stress [19].

The effects of stress and diet manipulation on forebrain dopamine in rats [19], along with data from studies utilizing combined genetic and behavioral approaches [20], suggest that highly palatable food consumption triggers an increase in dopamine secretion in the mesolimbic pathway from the ventral tegmental area (VTA) to the nucleus accumbens (Nacc). This activity further activates dopamine (DA) and opioid secretion from neurons throughout the homeostatic feeding network [21]. Specifically, the activation of DA receptor D2 (*Drd2*) in the VTA decreases the excitability of DA neurons and subsequent DA release in downstream DA cells, such as those located in the Nacc [22]. Decreased *Drd2* activity in the VTA allows for the increased activation of DA receptor D1 (*Drd1*) and increased extracellular DA in the Nacc. Taken together, this evidence raises the possibility that a highly palatable diet might mitigate the effects of early developmental stress by increasing mesolimbic dopaminergic activity and enhancing HPAa negative feedback efficiency via increased hippocampal glucocorticoid receptor expression.

### Current Study

Here, we investigated the impact of diet on the relationship between early-life stress and offspring behavioral phenotype, as well as candidate neural gene expression, in *Mus musculus*. Given the rewarding nature of a highly palatable Western-pattern diet (WPD) [20,23], our overarching hypothesis was that a WPD would mitigate the effects of maternal stress on offspring behavior and neural gene expression. Dams were chronically stressed during the last week of gestation (beginning on estimated gestation day 14). The diet of half of the dams was switched from a standard chow diet (SCD) to WPD on the first day of the stress treatment. We predicted that stressed dams fed SCD would spend less time with their pups and display an increased latency to retrieve displaced pups relative to NS dams and their WPD-fed counterparts. The latency to retrieve a displaced pup and the time a dam spent with her pups were used as indicators of maternal behavior and offspring early postnatal experience, with higher latency to retrieve and less time spent with offspring suggestive of early postnatal stress. Additionally, to account for the potential impact of the developmentally sensitive pubertal period on the relationship between early stress and diet, offspring were assigned to either SCD or WPD at the time of weaning (postnatal day 21). We hypothesized that prenatal and early postnatal stress would interact with maternal diet to predict offspring anxiety-like behavior measured using the open-field (OF) test. Behaviors observed in the OF, such as freezing, time spent exploring the inner zone (vs. the outer zone, an indication of thigmotaxis) of the open field apparatus, and general locomotor activity within the first 5 min of the test, are commonly used to assess anxiety-like behavior in rodents. We hypothesized that early stress would promote less exploratory (less time in the inner zone of the apparatus, greater distance traveled) and more anxiety-like (more time spent freezing, less distance traveled) behavior in the OF for the offspring of SCD-fed dams compared to NS controls. We predicted that the offspring of stressed dams who were fed a WPD would exhibit more exploratory, and less anxiety-like behavior compared to controls. We also hypothesized that a post-weaning WPD would further reinforce any interaction between maternal WPD and PS.

In addition to assessing behavior, we measured mRNA expression for genes encoding the glucocorticoid receptor, *Nr3c1*, and dopamine receptors *Drd1* and *Drd2*. We chose these genes because all three are implicated in stress- and reward-related behavior in brain regions that modulate stress reactivity and hedonic behavior: hippocampus, Nacc, and VTA. There is also evidence that the activation of VTA-Nacc DA projection neurons enhances phasic DA release via the activation of D1-medium spiny neurons (MSNs) and reduces phasic DA release via D2-MSN activity [24]. Moreover, the enhancement of Nacc D1-MSN activity is associated with resilience to the negative effects of chronic social defeat, whereas the enhancement of D2-MSNs is associated with the negative effects of chronic social defeat stress [25]. Therefore, we predicted that, compared to control offspring, the offspring of stressed dams who were fed a SCD would have lower *Nr3c1* expression in the hippocampus, resulting in more anxiety-like behavior due to less efficient negative regulation of the stress response. If WPD mitigates the effects of maternal stress on *Nr3c1* expression, then the offspring of stressed dams who were fed a WPD should exhibit *Nr3c1* expression levels and behavior similar to the control group. We also predicted that low levels of *Drd2* expression in the VTA would be associated with higher levels of *Drd1* expression in the Nacc among offspring of stressed dams fed a WPD, promoting more exploratory and less anxiety-like behavior, whereas higher levels of VTA *Drd2* would be associated with lower levels of Nacc *Drd1* among offspring of stressed dams fed a SCD, promoting less exploratory and more anxiety-like behavior.

## 2. Materials and Methods

### 2.1. Animals

Animals were female C57BL/6J mice (*n* = 49, mean age = 109.17 days, *SD* = 24.67 days) and their offspring (*n* = 244). All animals were housed in polycarbonate cages with Sanichip^®^ bedding and were provided with ad libitum access to food and water. The colony was maintained on a 12-h light–dark cycle (lights on at 0930 h). All animals were treated per the *Policy on Humane Care and Use of Laboratory Animals* and all procedures were approved by the Institutional Animal Care and Use Committee at Oklahoma State University (protocol # AS-14-1).

For breeding, females were placed in the home cage of a male for 3 days. After the breeding period, the females were removed from the male’s cage. Pregnant females were individually housed and provided with nesting material (cotton nestlets). Litters (*n* = 49) were composed of 4–9 pups (Mean = 5.02, *SD* = 1.73). Litters of dams that failed to rear four or more healthy pups to weaning (*n* = 4) or lost more than two pups that were born alive (*n* = 3) were excluded from analyses. Losses did not vary by maternal treatment (*r* = −0.04, *p* = 0.57).

Dams were weighed at parturition. There were no significant differences between dams fed a SCD (Mean = 23.12, SEM = 0.40) and dams fed a WPD (Mean = 22.64, SEM = 0.54). Offspring were weighed after completion of behavior experiments. Overall, males weighed more than females (*p* < 0.0001). WPD-fed males weighed significantly more than SCD-fed males (*p* < 0.0001), but there were no significant weight differences in females (*p* = 0.09) (Figure 1).

### 2.2. Experimental Design and Treatment Groups

We used a full factorial design with four maternal treatment groups and two offspring diet treatments per sex within each maternal treatment, resulting in a total of 16 offspring groups (Figure 2). Pregnant females in the prenatal stress (PS) treatment (*n* = 24) underwent restraint stress in a ventilated plastic tube (Kaytee Critter Trails Fun-nels, 6.35 cm diameter, 8.85 cm long) that was placed inside the home cage. A light (36,563 lux) was placed directly above the home cage. Mice were left in the restraint apparatus under the bright light for 45 min per day between the hours of 1100 and 1400, during approximate gestation days 14-20. Dams in the no stress (NS) treatment (*n* = 25) were not manipulated in any way.

Dams in the WPD groups (stress *n* = 12, no stress *n* = 12) had their diet changed from SCD (Laboratory Rodent Diet 5001, LabDiet, St. Louis, MO, USA) to WPD (D12079B, Research Diets, New Brunswick, NJ, USA; Figure 3) on estimated gestation day (GD) 14 (actual Mean GD = 13.20, *SD* = 1.27). Dams who were assigned to the SCD groups (stress *n* = 12, no stress *n* = 13) had no change in their diet.

Pups were weaned on postnatal day (PND) 21. Half of each litter was randomly assigned to a SCD and the other half to a WPD. Pups were housed with same-sex/same-diet siblings.

### 2.3. Maternal Behavior

Home-cage activity was used as a proxy for time dams spent in the nest, with higher activity used as an indication of dams spending less time in the nest with their pups. Dam activity levels were monitored continuously from parturition for approximately 24 h (the beginning of the first light cycle after parturition until the start of the second light cycle after parturition) using an automated monitoring system that recorded the number of infrared beam breaks per minute (VitalView Animal Monitoring Software, Version 5.0).

Pup retrieval was measured in the first 24 h after parturition. The dam was removed from the home cage for 2 min. During this time, one pup was removed from the nest and placed in the far corner of the home cage. The dam was then returned to the home cage, and pup retrieval was measured as the latency, in seconds, for the dam to pick up the pup and return it to the nest.

### 2.4. Offspring Behavior

Offspring completed an OF for measurement of anxiety- and exploratory-like behavior in adulthood (between PND 80 and 83). The open-field arena comprised a 16-square grid enclosed by a clear Plexiglass box (60.96 × 60.96 × 60.96 cm) with no top. Trials were conducted during the light cycle between 1000 and 1400 h. At the start of each trial, the mouse was placed in the center of the OF in an opaque PVC cylinder. The cylinder was removed after ~15 s, and the trial was run for 5 min, starting when the cylinder was lifted. The OF was thoroughly cleaned with 70% ethanol and given 5 min to dry before each trial began. Each trial was video recorded and scored using Any-Maze software (Stoelting Co., Wood Dale, IL, USA). Three parameters were measured: total distance traveled, time spent freezing, and time spent in the inner zone of the OF (middle four squares). The time spent in the outer zone of the OF (12 squares surrounding center) is not included, since it is simply the inverse of time spent in the inner zone. Lower locomotor activity (measured via distance traveled), less time spent in the inner zone of the OF, and more time spent freezing are suggestive of higher levels of anxiety [26].

### 2.5. Neural Tissue Collection

Adult offspring (PND 80–83) were sacrificed by cervical dislocation after behavioral testing. Brains were extracted into RNAlater, stabilized at 4 °C for 24–48 h, and stored at −20 °C until microdissection. Target brain regions (Nacc, hippocampus, and VTA) were dissected into RNAlater with a scalpel under a dissecting microscope [27]. Brain structures were identified using the mouse brain atlas of Paxinos and Franklin [28].

### 2.6. RNA Extraction and Quantitative Real-Time PCR

RNA was extracted from frozen tissue using the RNeasy Mini Kit (Qiagen, Hilden, Germany) according to the manufacturer’s protocol. RNA purity and concentration were evaluated using a NanoDrop 2000 spectrophotometer. Complementary DNA (cDNA) was synthesized the same day using iScript RT Supermix (Bio-Rad Laboratories, Hercules, CA, USA). Quantitative PCR (qPCR) for the genes of interest (*Drd1*, *Drd2*, and *Nr3c1*) and the reference gene β-actin (*Actb*) was performed using the SYBR Green PCR Master Mix (Bio-Rad) and CFX Connect Real-Time PCR Detection System (Bio-Rad). Each reaction was run in triplicate. Primer sequences for all genes are from [29]. Product specificity was assessed by an analysis of melting curves; the amplification efficiency was 90% to 110% for each primer pair. The results were analyzed by the ΔΔCt method and normalized to the expression of β-actin.

## 3. Results

### 3.1. Maternal Behavior

Two separate two-way analysis of variance (ANOVA) tests were conducted to examine the effects of maternal diet and PS on two maternal behaviors: the amount of time dams spent outside of the nest (i.e., away from pups), and pup retrieval. The Šidák correction was used to account for multiple comparisons of maternal behavior.

#### 3.1.1. SCD-Fed Stressed Dams Spend Less Time in Nest with Pups Than WPD-Fed Stressed Dams and Non-Stressed Dams

There was a significant interaction effect of PS and maternal diet on the amount of time dams spent outside the nest in the first 24 h after their pups were born (F (1, 45) = 7.67, *p* = 0.01). Post hoc comparisons indicated that PS dams who were fed SCD spent significantly more time outside the nest as compared to NS dams who were fed SCD (the control group) (*t* (45) = 3.07, *p* = 0.01; Figure 4A). In contrast, time outside of the nest did not differ between PS and NS dams that were fed WPD.

There was a significant interaction effect of PS and maternal diet on the amount of time dams spent outside the nest in the first 24 h after their pups were born (F (1, 45) = 7.67, *p* = 0.01). Post hoc comparisons indicated that PS dams who were fed SCD spent significantly more time outside the nest as compared to NS dams who were fed SCD (the control group) (*t* (45) = 3.07, *p* = 0.01; Figure 4A). In contrast, time outside of the nest did not differ between PS and NS dams that were fed WPD.

#### 3.1.2. Stressed Dams Display a Higher Latency to Retrieve Displaced Pups, but a WPD Significantly Decreases Latency to Retrieve Pups

A similar pattern was observed for the pup retrieval test. PS and maternal diet interacted to exert a significant effect on pup retrieval in the first 24 h postpartum (F (1, 45) = 5.06, *p* = 0.03), such that the mean latency to retrieve a displaced pup was significantly higher for SCD-fed and PS dams compared to the NS control group (*t* (45) = 3.57, *p* = 0.002; Figure 4B). Retrieval latency was statistically equivalent in PS vs. NS WPD-fed dams.

### 3.2. Open-Field Behavior in Adult Offspring

Open-field (OF) behaviors were analyzed using linear mixed models (LMM) with the Satterthwaite approximation. Animals whose diets changed at weaning and those whose diets remained consistent were analyzed separately. In both analyses, PS, maternal diet, and sex were included as fixed effects, and litter ID as a random effect. When there were significant main or interaction effects of sex, males and females were analyzed separately. Post hoc pairwise comparisons were run with one-way ANOVAs using the Bonferroni correction for multiple comparisons.

#### 3.2.1. The Offspring of WPD-Fed Stressed Dams Exhibit More Locomotor Activity Than the Off-Spring of SCD-Fed Stressed Dams

There were significant interaction effects for sex and PS (F (1, 104) = 18.99, *p* < 0.001), as well as sex and maternal diet (F (1, 104) = 55.58, *p* < 0.001) on the distance traveled by animals whose diet remained consistent at weaning (Figure 5A,B). For male offspring whose diets remained consistent, there was a significant main effect of stress (*t* (25.25) = 12.22, *p* < 0.0001) and no main or interaction effects for maternal diet. Specifically, maternal stress significantly reduced locomotor activity, regardless of diet (Figure 5A).

Overall, females were more active than males (*t* (98.95) = 11.51, *p* < 0.0001). The female offspring of PS dams displayed less locomotor activity than the offspring of NS dams (*t* (17.65) = 16.03, *p* < 0.001). However, female PS+WPD offspring displayed more locomotor activity than female PS+SCD offspring (*t* (18.81) = 8.49, *p* < 0.001). Likewise, female NS+WPD offspring displayed more locomotor activity than the female NS+SCD (*t* (16.63) = 5.27, *p* = 0.0004).

Male NS+SCD^+^ (see Figure 2 for notation definitions) were more active than their PS+SCD^+^ counterparts (*t* (36.54) = 6.22, *p* < 0.00001) (Figure 5C). Whereas male PS+SCD^+^ were less active overall than their NS+SCD^+^ counterparts (*t* (26.98) = −6.29, *p* < 0.00001), PS+WPD^+^ males were more active than PS+SCD^+^ (*t* (19.63) = 3.35, *p* = 0.02).

Female offspring of NS dams who had their diets switched at weaning were significantly more active than their counterparts in the PS groups (*t* (30.66) = 25.31, *p* < 0.00001) (Figure 5D). Interestingly, female NS+SCD^+^ were significantly less active than their NS+WPD^+^ counterparts (*t* (27.19) = −7.26, *p* < 0.00001).

#### 3.2.2. The WPD-Fed Offspring of Stressed Dams Spend More Time in the Inner Zone of the OF than the SCD-Fed Offspring of Stressed Dams When Diet Remains Consistent, but a Diet Change at Weaning Reduces This Effect

There was a significant three-way interaction between PS, maternal diet, and sex on the amount of time animals whose diets remained consistent spent in the inner zone of the OF (F (1, 104) = 21.5, *p* < 0.001). Male (*t* (25.3) = −11.5, *p* < 0.001) and female (*t* (17.6) = −5.44, *p* < 0.001) offspring of PS dams spent significantly less time in the inner zone of the OF than their NS counterparts (Figure 6A-B). However, the WPD-fed male (*t* (22.4) = 5.60, *p* < 0.001) and female (*t* (18.8) = 17.64, *p* < 0.001) offspring of PS dams spent significantly more time in the inner zone of the OF than their SCD-fed counterparts (Figure 6A,B). There was no significant effect of diet on the amount of time spent in the inner zone of the OF among the offspring of NS dams (Figure 6A,B).

For animals whose diets were switched at weaning, there was a significant three-way interaction between PS, maternal diet, and sex on the amount of time spent in the inner zone of the OF (F (1, 129) = 24.68, *p* < 0.001). Overall, the offspring of PS dams spent less time in the inner zone of the OF compared to the offspring of NS dams (*t* (37.3) = 19.9, *p* < 0.001) (see Figure 6C,D). Male NS+SCD^+^ spent significantly more time in the inner zone of the OF than their NS+WPD^+^ counterparts (*t* (33.4) = 6.19, *p* < 0.001). Although the male offspring of PS dams spent less time in the inner zone of the OF overall, male PS+WPD^+^ spent significantly more time in the inner zone than their PS+SCD^+^ counterparts (*t* (21) = −3.67, *p* = 0.009). Alternatively, there was no effect of diet on the female offspring of PS dams. However, female NS+SCD^+^ spent significantly more time in the inner zone of the OF than their NS+WPD^+^ counterparts (*t* (27.2) = 4.92, *p* < 0.001).

#### 3.2.3. The Offspring of WPD-Fed Stressed Dams Spend Less Time Freezing Than the Offspring of SCD-Fed Stressed Dams, a Diet Change at Weaning Reverses This Effect in Females

For animals whose diets remained consistent at weaning, there was a significant interaction between PS and maternal diet on the amount of time animals spent freezing (immobile for ≥1 s) in the OF (F (1, 104) = 384.51, *p* < 0.001). There were no main or interaction effects of sex.

Maternal diet had opposing effects on the freezing behavior of offspring of PS and NS dams (*t* (32.3) = −26.2, *p* < 0.001). PS+WPD offspring spent less time freezing than PS+SCD offspring (*t* (28.4) = −22.15, *p* < 0.001; Figure 7A). In contrast, NS+WPD offspring spent slightly but significantly more time freezing than NS+SCD offspring (*t* (36.4) = 5.97, *p* < 0.001).

For animals whose diets were switched at weaning, there was a significant three-way interaction between PS, maternal diet, and sex on the amount of time spent freezing (F (1, 129) = 31.42, *p* < 0.001). Overall, the PS offspring spent significantly more time freezing than their NS counterparts (*t* (37.3) = 10.4, *p* < 0.001). Notably, in the female offspring of PS dams, a diet change at weaning reversed this effect of maternal diet, such that PS+WPD^+^ offspring spent more time freezing than PS+SCD^+^ offspring (*t* (33.7) = 3.99, *p* = 0.002; Figure 7C). In contrast, the reduction in time spent freezing in male PS+WPD vs. PS+SCD offspring remained, regardless of post-weaning diet (*t* (19.6) = 4.44, *p* < 0.001).

### 3.3. Neural Gene Expression

Expression data were analyzed with separate three-way ANOVAs for offspring whose diets remained consistent and those whose diets changed at weaning. We treated NS+SCD^-^ as controls. For each gene, we tested for effects of PS, maternal diet, sex, and their interactions on normalized expression levels; figures show Log2 fold change relative to controls. Male and female data were pooled unless a significant effect of sex was detected. Post hoc tests were run with one-sample *t*-tests and the Šidák correction for multiple comparisons.

#### 3.3.1. The Effects of PS on *Drd1* Expression in the Nucleus Accumbens Depend on Both Maternal and Post-Weaning Diet, and Are More Pronounced in Female Offspring

There were significant three-way interactions between PS, maternal diet, and sex on *Drd1* expression for both groups (diet consistent: F (1, 104) = 21.17, *p* < 0.001; diet changed: F (1, 129) = 563, *p* < 0.001). Therefore, analyses were split by sex.

The significant interaction between maternal diet and PS remained for both male and female offspring whose diet was consistent at weaning (males: F (1, 62) = 1118, *p* < 0.0001; Figure 8A; females: F (1, 45) = 1401, *p* < 0.0001; Figure 8B), and whose diet was changed at weaning (males: F (1,62) = 11.26, *p* = 0.0014; Figure 8C; females: F (1,67) = 231.90, *p* < 0.0001; Figure 8D). Notably, *Drd1* expression was significantly elevated relative to same-sex controls among PS+WPD^−^ offspring and reduced among PS+SCD^−^ offspring (males, WPD: *t* (14) = 32, *p* < 0.0001; SCD: *t* (18) = 27.12, *p* < 0.0001; females, WPD: *t* (9) = 9.96, *p* < 0.0001; SCD: *t* (10) = 34.44, *p* < 0.0001). There was also a small but significant increase in *Drd1* expression in female NS+WPD^-^ offspring (*t* (9) = 9.96, *p* < 0.0001).

When combined with a diet change at weaning, the effects of maternal stress and diet were sex-specific. In the male offspring of PS dams, the effect of maternal diet on *Drd1* expression was effectively canceled out by the diet change (i.e., Figure 8A vs. Figure 8C). The effect of diet change at weaning was even more pronounced in female offspring of PS dams, resulting in the reversal of the effects of maternal diet on *Drd1* expression as compared to females whose diet matched their mothers’ (i.e., Figure 8B vs. Figure 8D). Expression differences relative to female controls remained significant for both maternal diet groups (WPD: *t* (15) = 16.87, *p* < 0.0001; SCD: *t* (17) = 27.43, *p* < 0.0001; Figure 8D), but the direction of change in *Drd1* expression was reversed as compared to females whose diet matched their mothers’ (i.e., Figure 8B vs. Figure 8D).

NS+SCD^+^ offspring had significantly higher *Drd1* expression relative to controls. In contrast to the PS groups, this effect of diet was more pronounced in males (*t* (16) = 31.28, *p* < 0.0001; Figure 8C) than in females (*t* (23) = 8.13, *p* < 0.0001; Figure 8D).

#### 3.3.2. The Offspring of Stressed Dams Have Decreased *Drd2* Expression in the Ventral Tegmental Area When Exposed to a Maternal and/or Postweaning WPD, and Increased Expression When Maternal and Postweaning Diets Are SCD

As for *Drd2*, there was a significant interaction between maternal diet and PS in offspring whose post-weaning diet matched their mothers’ (F (1, 104) = 3039.29, *p* < 0.001), but no effect sex. For offspring whose diet changed at weaning, there were main effects of PS (F (1, 129) = 1165.58, *p* < 0.001) and maternal diet (F (1, 129) = 139.40, *p* < 0.001), but no effect of sex and no interaction effects.

The direction of the effect of maternal stress on offspring *Drd2* expression depended on maternal diet when post-weaning diet was unchanged (Figure 9A). Expression was significantly increased relative to controls in the PS+SCD^-^ mice (*t* (24) = 45.34, *p* < 0.0001) and decreased in PS+WPD^-^ mice (*t* (24) = −4.64, *p* = 0.0001). In contrast, PS+SCD^+^ mice had significantly decreased *Drd2* expression (*t* (36) = 34.17, *p* < 0.0001; Figure 9B), which was comparable to PS+WPD^+^ mice (Figure 9A). Finally, a switch to SCD at weaning resulted in a lesser, albeit still significant, reduction in *Drd2* expression in the PS+WPD offspring (*t* (30) = 19.15, *p* < 0.0001; Figure 9B).

In the absence of maternal stress, *Drd2* expression was slightly but significantly elevated relative to controls in offspring exposed to a maternal, but not post-weaning, WPD (Maternal WPD: *t* (24) = 4.63, *p* = 0.0001).

#### 3.3.3. The Offspring of WPD-Fed Stressed Dams Have Increased Expression of Glucocorticoid Receptor, *Nr3c1*, in the Hippocampus

There was a significant interaction between maternal diet and PS on *Nr3c1* expression in offspring whose post-weaning diet remained consistent with their mothers’ (F (1, 44.70) = 10.24, *p* = 0.003), but no main or interaction effects of sex. For offspring whose diet was changed at weaning, there was a moderately significant three-way interaction between PS, maternal diet, and sex (F (1, 127.4) = 4.26, *p* = 0.04) (Figure 10A). When split by sex, the interaction between PS and diet remained significant for both sexes (males: F (1, 31.24 = 7.71, *p* = 0.009; females: (F (1, 31.88) = 24.44, *p* < 0.0001) (Figure 10B,C). Overall, *Nr3c1* expression was significantly higher in PS+WPD offspring relative to controls (no diet change: *t* (33) = 3.24, *p* = 0.003); diet change, males: *t* (14) = 2.52, *p* = 0.02; diet change females: *t* (15) = 4.48, *p* = 0.0004; see Figure 10). In contrast, expression was significantly reduced in the PS+SCD offspring, but only when diet was switched to a WPD at weaning (males: *t* (37) = 4.19, *p* = 0.0002; females: *t* (17) = 6.24, *p* < 0.0001).

## 4. Discussion

In the present set of experiments, we studied the impact of PS and diet on the stress-related behavior and neural gene expression of adult offspring. Overall, we found that the consequences of early-life stress for adult behavior and neural gene expression are significantly impacted by diet.

The offspring of SCD-fed PS dams who remained on a SCD post-weaning displayed reduced Nacc *Drd1* and hippocampal *Nr3c1* expression, increased VTA *Drd2*, and increased anxiety-like behavior compared to NS mice. This result is consistent with previous work, which demonstrated that prenatal restraint stress exposure results in decreased hippocampal *Nr3c1* expression in offspring, together with long-lasting changes in DA sensitivity in the Nacc of adult offspring [30,31].

In contrast, the offspring of PS, WPD-fed dams who were fed a WPD post-weaning displayed increased *Drd1* and *Nr3c1* expression and decreased *Drd2* expression compared to their SCD-fed counterparts, along with a more exploratory-like behavioral phenotype. Our results are consistent with others that have found an increase in exploratory-like behavior among the offspring of dams that were fed a cafeteria or high-fat diet [32,33], although dams were not subjected to PS in those studies, and offspring were provided with standard SCD at weaning. In contrast to our findings, there have also been several studies that demonstrate that a high-fat perinatal diet can lead to increased anxiety-like behavior in offspring [34,35]. Importantly, evidence suggests that the timing of high-fat or cafeteria diet administration plays a role in the development of offspring anxiety-like behavior. Elegant experiments conducted by Wright and colleagues [36] demonstrated that administering a WPD several weeks before conception leads to significantly higher incidences of maternal obesity and generally increases anxiety-like behavior in adult offspring. However, restricting the high-energy diet to gestation and lactation led to decreased anxiety-like behavior in offspring. Indeed, our results are consistent with studies that limit high-energy diets to gestation and lactation, preventing maternal obesity, and demonstrating a protective effect of these modified diets on anxiety-like adult behavior [32,36,37]. Moreover, the behavioral and gene expression differences we observed in the adult offspring of PS, WPD-fed dams who were maintained a WPD post-weaning, compared to their SCD-fed counterparts, indicates that stress-related reductions in dopamine signaling may be mitigated by diet [19,37].

Interestingly, the offspring of PS dams whose diets remained consistent at weaning displayed a marked imbalance between Nacc *Drd1* and VTA *Drd2* expression, such that those offspring fed SCD had higher *Drd2* and lower *Drd1* expression, whereas those fed a WPD had higher *Drd1* and lower *Drd2* expression. Recent research demonstrates that this type of *Drd1/Drd2* imbalance could indicate the sensitization of the mesolimbic system and contribute to addictive-like behavior [38]. This has potentially important implications for human health. For example, the offspring of mothers who experience chronic stress while pregnant and consuming a WPD may develop an increased susceptibility to engage in both compulsive eating and problematic drug use. Indeed, there is evidence that VTA *Drd2* knockout rats demonstrate an increase in incentive motivation toward cocaine and highly palatable food [39], and that a high-fat/high-sugar diet enhances the abuse-related effects of cocaine [40,41]. Taken together, this evidence further supports the important role of *Drd2* in compulsive eating and addictive behavior.

The differences in behavior and gene expression we observed in the offspring of WPD-fed mothers could be due to a direct effect of the diet, as milk composition reflects the composition of the WPD. Maternal diet may also impact offspring indirectly via effects on maternal behavior. Both under- and over- nutrition have been shown to change maternal behavior [42,43], consistent with the observations of maternal behavior in the current study. Increased maternal care is also associated with a reduction in neuroendocrine responses to stress in rodent offspring that persists throughout life [44,45]. Though there was a significant effect of diet on maternal behavior in the current study, it is unlikely that the behavioral and neural gene expression effects observed in offspring were due exclusively to an indirect effect of maternal behavior, since changing diet at weaning also changed behavior and gene expression profiles. Instead, perhaps changing offspring diet at weaning serves as an environmental stressor capable of inducing stress-related gene expression changes and behavior. Moreover, the sex differences observed in the animals whose diets were changed indicate that females may be more sensitive to the stressor of diet change at weaning than males. For example, changing diet at weaning also ameliorated the increase in exploratory behavior observed in S+WPD^-^ females. Indeed, both males and females in these groups displayed more anxious-related behavior than controls, as indicated by increased freezing, less distance traveled, and less time in the center of the OFT, which is consistent with studies that have assessed anxiety-related behavior in the offspring of high-fat-diet-fed dams that were subsequently maintained on SCD [34,43,44]. However, male S+WPD^+^ displayed less anxiety-like behavior compared to S+SCD^+^ males. This is consistent with a large body of literature suggesting that males are more sensitive to stress during the prenatal and early postnatal period (for review, see [46]). While females appear to display resiliency to the effects of prenatal and early postnatal stress, the results of these developmental insults on phenotype are often exposed following periods of hormonal activation and fluctuation, such as during the adolescent period. Future work should explore the mechanisms contributing to sex differences in the context of early stress and dietary changes.

## 5. Conclusions

The results of this study add to existing evidence that the interaction between stress and diet can have long-term effects on behavior [17,32,34,36,43,47,48,49,50,51,52,53,54,55] and suggest that a WPD may have the potential to mitigate stress-induced gene expression changes that occur early in development and the anxiety-like behavioral phenotypes such molecular changes lead to. This is not to suggest that consuming highly palatable comfort food is a healthy or desirable form of stress management. The neurobiological processes that occur during stress and feeding contribute to sustained negative reinforcement, leading to a short-term gain in well-being (stress reduction) with a long-term cost to health (i.e., obesity, metabolic disease) [56]. Instead, these results could help explain why certain individuals continue over-consuming highly palatable comfort food to their detriment, as well as why non-surgical weight-loss interventions have such a low success rate in the long term [57]. Although physiology plays a major role in the difficulties of long-term weight-loss [57], at least part of the story may concern the impact of a maternal WPD on a stressed and developing brain.

## Figures and Tables

**Figure 1 ijms-23-09245-f001:**
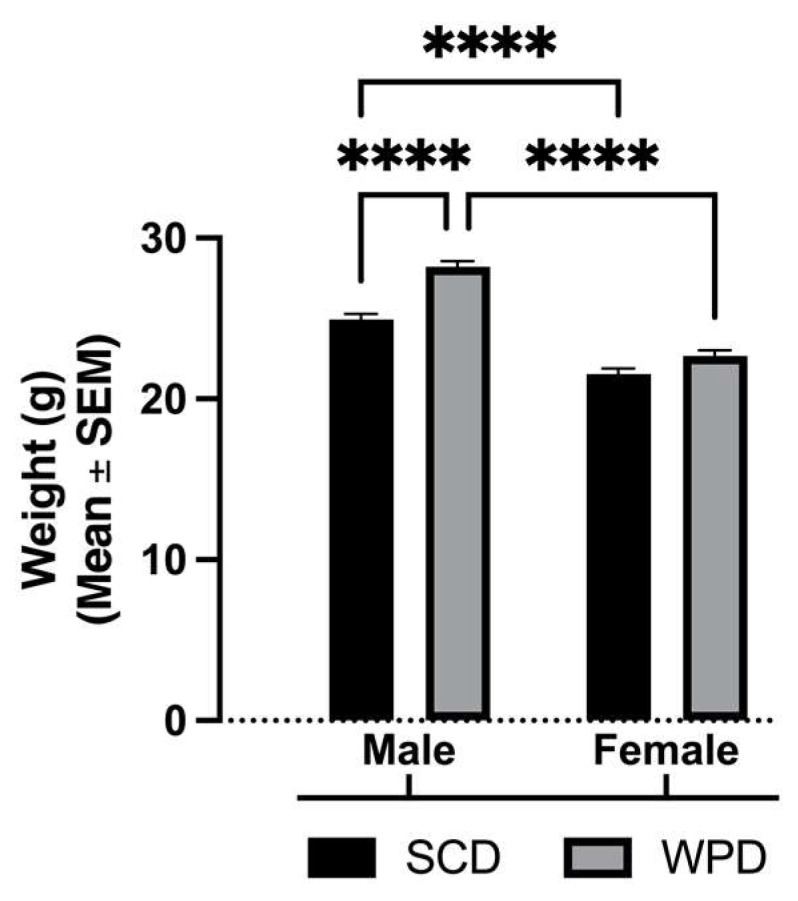
Adult weights (in grams) of offspring by diet and sex. SCD = standard chow diet, WPD = western-pattern diet. **** *p* < 0.0001.

**Figure 2 ijms-23-09245-f002:**
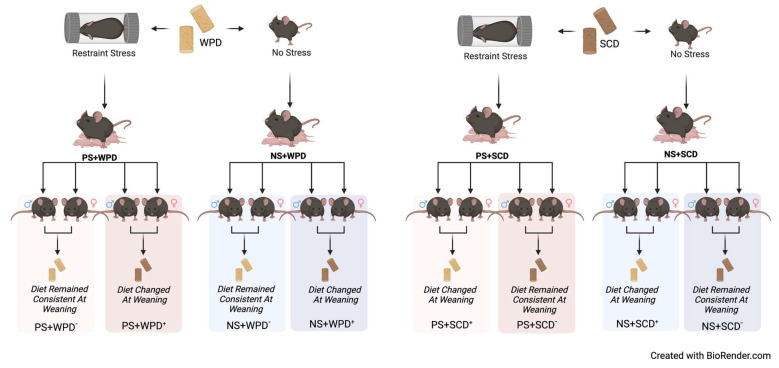
Graphic representation of experimental groups. WPD: Western-pattern diet; SCD: standard chow diet; PS: Prenatal Stress; NS: No Stress; (−) indicates no diet change at weaning; (+) indicates diet change at weaning. Figure created with Biorender.com.

**Figure 3 ijms-23-09245-f003:**
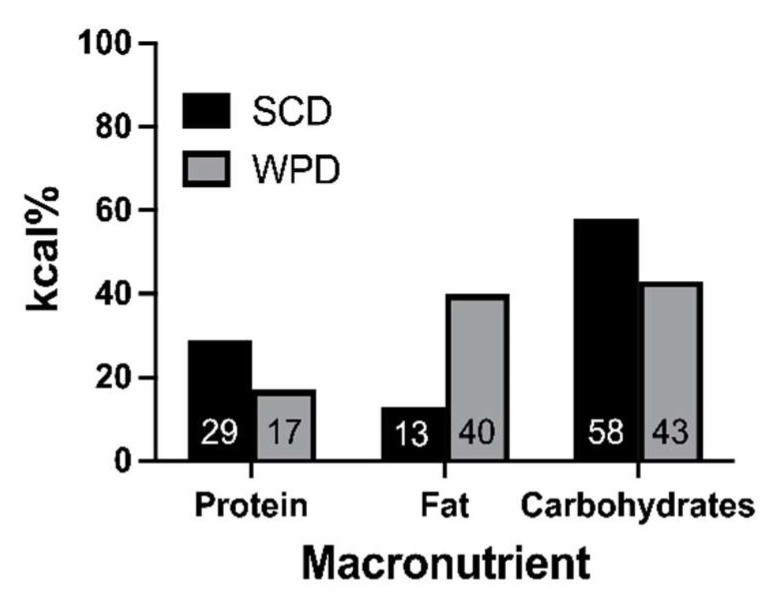
Macronutrient kcal/gram % of experimental (WPD) and control (SCD) diets. WPD = western pattern diet; SCD = standard chow diet.

**Figure 4 ijms-23-09245-f004:**
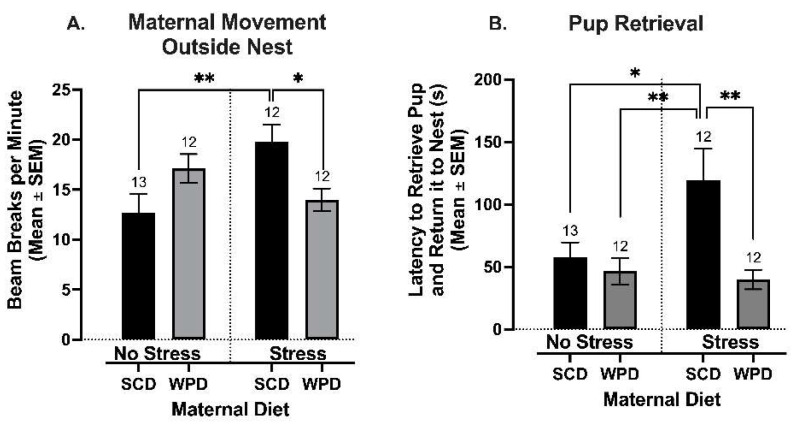
Maternal behavior represented by (**A**) the amount of time the dam spent moving around outside of the nest and (**B**) the amount of time it took for a dam to retrieve a displaced pup and return it to the nest. Group *n*’s located above bars. WPD = Western pattern diet. SCD = standard chow diet. * *p* < 0.05, ** *p* < 0.01.

**Figure 5 ijms-23-09245-f005:**
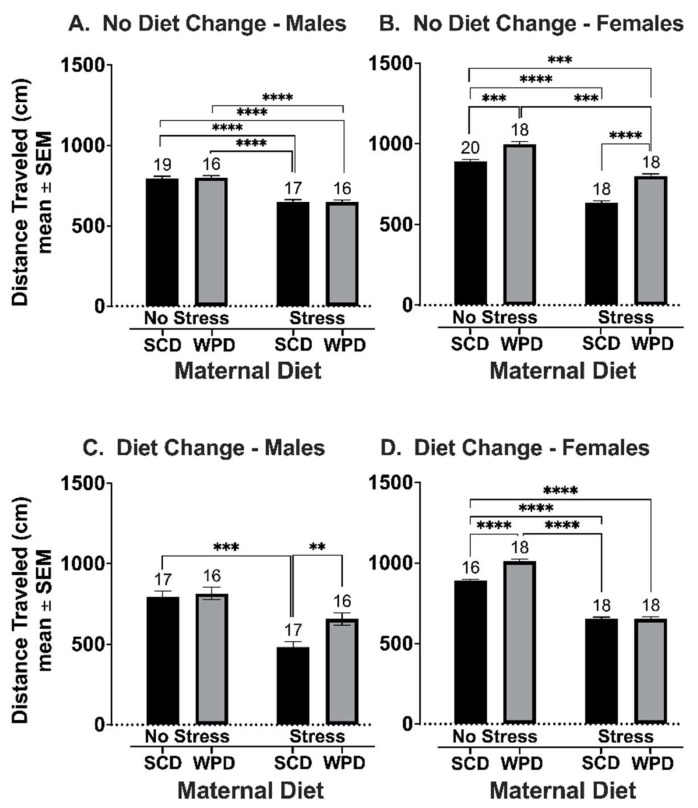
Total distance traveled in the OF for (**A**) males whose diets remained consistent at weaning; (**B**) females whose diets remained consistent at weaning; (**C**) males whose diets were switched at weaning; and (**D**) females whose diets were switched at weaning. Group *n*’s located above bars. WPD = western pattern diet; SCD = standard chow diet. ** *p* < 0.01; *** *p* < 0.001; **** *p* < 0.0001.

**Figure 6 ijms-23-09245-f006:**
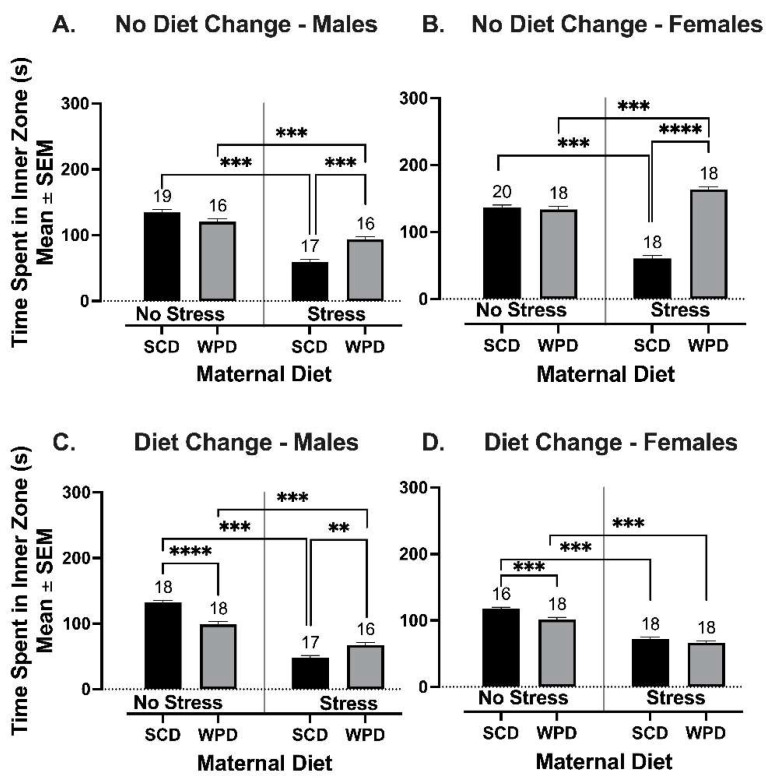
Time spent in the inner zone of the OF for (**A**) males whose diets remained consistent at weaning; (**B**) females whose diets remained consistent at weaning; (**C**) males whose diets were switched at weaning; and (**D**) females whose diets were switched at weaning. Group *n*’s located above bars. WPD = western pattern diet. SCD = standard chow diet. ** *p* < 0.01, *** *p* < 0.001; **** *p* < 0.0001.

**Figure 7 ijms-23-09245-f007:**
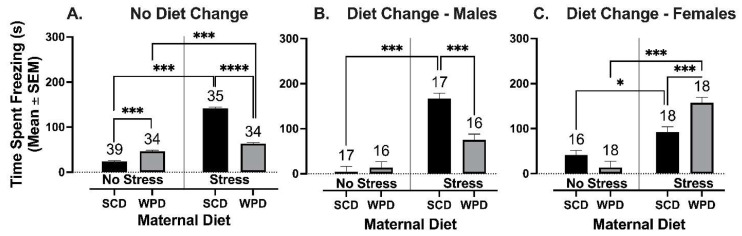
Total time that mice spent freezing during the OF. Freezing behavior was defined as lack of movement for ≥1 s. (**A**) Animals whose diets remained consistent at weaning. Data from male and female mice are pooled due to no significant main or interaction effects of sex. (**B**) Male mice whose diets were switched at weaning. (**C**) Female mice whose diets were switched at weaning. Group *n*’s located above bars. WPD = western pattern diet; SCD = standard chow diet. * *p* < 0.05, *** *p* < 0.001, **** *p* < 0.0001.

**Figure 8 ijms-23-09245-f008:**
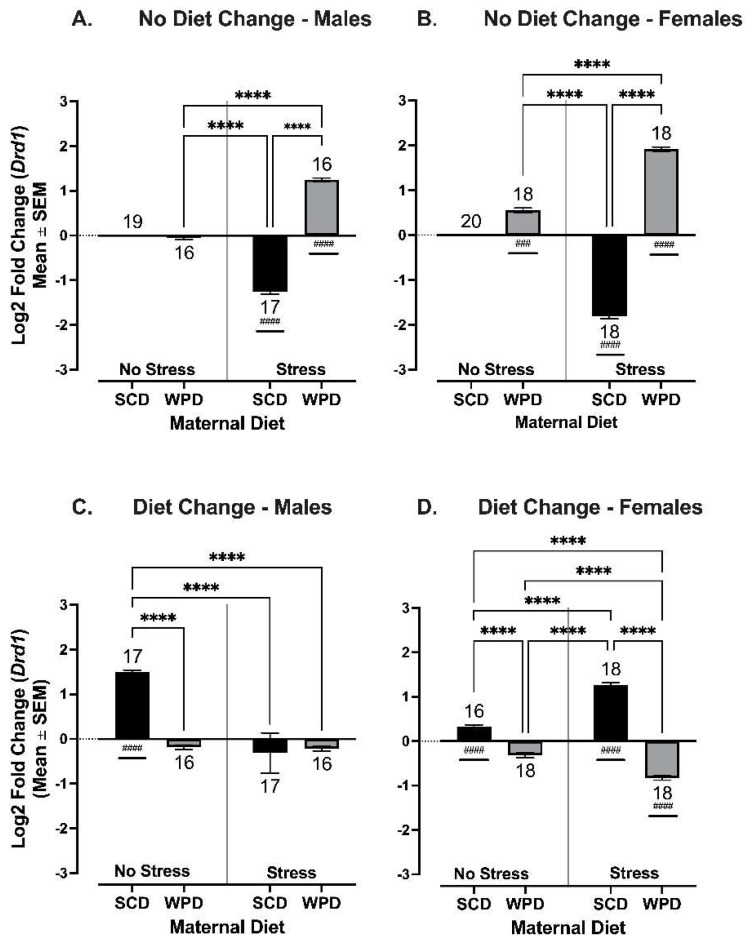
*Drd1* log2 fold change in expression in Nacc compared to no stress/SCD fed controls for (**A**) males whose diets remained consistent at weaning, (**B**) females whose diets remained consistent at weaning, (**C**) males whose diets were switched at weaning, and (**D**) females whose diets were switched at weaning. Group *n*’s located above/below bars. WPD = Western pattern diet, SCD = standard chow diet. **** *p* < 0.0001. #: significant fold change relative to control. ### *p* < 0.001, #### *p* < 0.0001.

**Figure 9 ijms-23-09245-f009:**
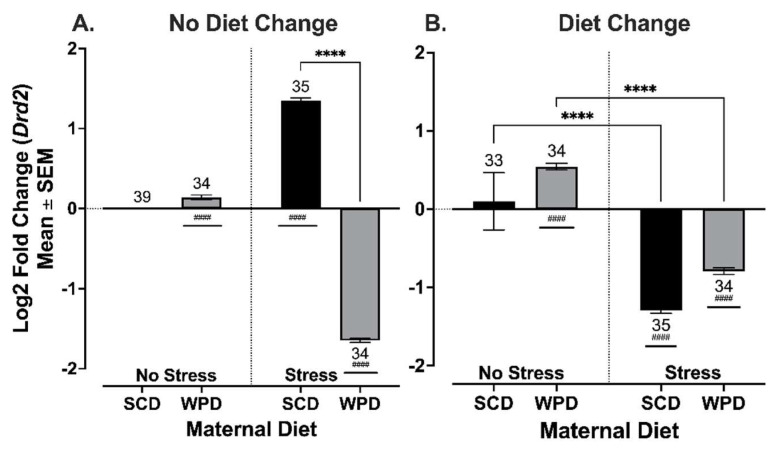
*Drd2* log2 fold change in expression in VTA compared to no stress/SCD-fed controls in (**A**) animals whose diets remained consistent at weaning, and (**B**) animals whose diets were switched at weaning. Sex was pooled for analysis. Group *n*’s located above/below bars. WPD = western pattern diet, SCD = standard chow diet. **** *p* < 0.0001. # Significant fold change from control. #### *p* < 0.0001.

**Figure 10 ijms-23-09245-f010:**
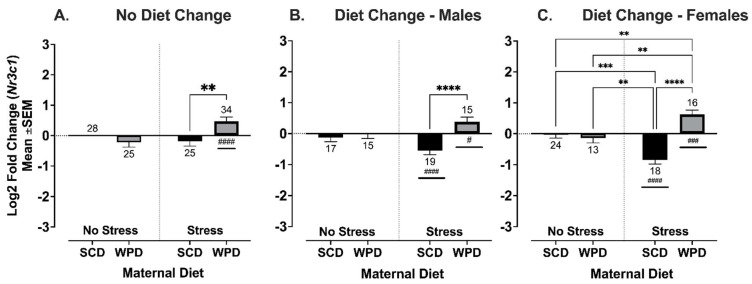
Log2 fold change of hippocampal glucocorticoid receptor, *Nr3c1*, compared to no stress/SCD fed controls. Group *n*’s located above/below bars. WPD = Western pattern diet, SCD = standard chow diet. ** *p* < 0.01, *** *p* < 0.001, **** *p* < 0.0001. #: significant fold change relative to control., # *p* < 0.01, ### *p* < 0.001, #### *p* < 0.0001.

## Data Availability

Data are available upon request from the corresponding author.

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
