# Peer review of "The Modification of Offspring Stress-Related Behavior and the Expression of Drd1, Drd2, and Nr3c1 by a Western-Pattern Diet in Mus Musculus"

_ijms, 2022, doi:10.3390/ijms23169245_

Round 1

Reviewer 1 Report

The manuscript entitled “The Modification of Offspring Stress-Related Behavior and Gene Expression by Diet ” by Clauss and coworkers is focusing an interesting and actual topic related to the investigation of interactions between western-pattern diet and prenatal stress and sex. Authors have focused on the impact the expression of anxiety-like behavior as well as the expression of the glucocorticoid receptor in the hippocampus, D1 dopamine receptors in the nucleus accumbens, and D2 dopamine receptors in the ventral tegmental area. The overall quality of this work is good and meets the basic requirement of the journal. The figures look promising and well organized. Literature searching and citation papers are supporting the main content.

Minor: 

  • I suggest introducing some abbreviation for standard chow instead of chow; in figures should be for instance STD and WDP, because WDP is also a chow.

  • Please add n for each experimental group in the figure captions.

  • Table with diet composition and caloric content might be useful for readers.

  • Did Authors collect data related to animal weights? or food intake?

Author Response

Point 1: I suggest introducing some abbreviation for standard chow instead of chow; in figures should be for instance STD and WDP, because WDP is also a chow.

Response 1: Great suggestion! I changed all mention of standard chow to “STC” in 

Point 2: Please add n for each experimental group in the figure captions.

Response 2: Done.

Point 3: Table with diet composition and caloric content might be useful for readers.

Response 3: Added table with diet composition.

Point 4: Did Authors collect data related to animal weights? or food intake?

Response 4: We collected data on weight but not food intake. I added information on weight in the Animals section.

Reviewer 2 Report

The authors have carried out an interesting research work, in which they relate the mother's diet and the stress suffered during pregnancy with the future behaviour of the offspring.

However, I believe that for the manuscript to be published, some changes need to be made.

It should be indicated from the beginning of the paper that it is a study carried out with experimental animals; it should even appear in the title.

I have checked the publication guidelines and they specify the different sections that should appear, but the order of these is not indicated. If possible, it would be highly recommended that the materials and methods section precedes the results and discussion. This would make the study much easier to understand. 

In the results section, the authors should change the way of naming the different experimental groups, as the incessant repetition of the same terms makes it very difficult to understand the results. As a suggestion, I propose a scheme or diagram showing the different study groups coded in a clear way that makes it possible to follow the text.

It would also be interesting to know the exact composition of the diet fed to the animals.

I look forward to hearing from you.

Yours sincerely

Author Response

Point 1: It should be indicated from the beginning of the paper that it is a study carried out with experimental animals; it should even appear in the title.

Response 1: I have changed the title to include this.

Point 2: I have checked the publication guidelines and they specify the different sections that should appear, but the order of these is not indicated. If possible, it would be highly recommended that the materials and methods section precedes the results and discussion. This would make the study much easier to understand. 

Response 2: This has been done.

Point 3: In the results section, the authors should change the way of naming the different experimental groups, as the incessant repetition of the same terms makes it very difficult to understand the results. As a suggestion, I propose a scheme or diagram showing the different study groups coded in a clear way that makes it possible to follow the text.

Response 3: I have made an attempt at this, I hope it is satisfactory.

Point 4: It would also be interesting to know the exact composition of the diet fed to the animals.

Response 4: A figure with macronutrient content has been added. References lead to more detailed micronutrient information.

Reviewer 3 Report

The authors investigate Offspring Stress-Related Behavior and Gene Expression modifications in a mouse model.

In general the manuscript is well written. One area that needs to be addressed is that sex differences were assessed in the offspring, but this is not reflected in the title or abstract. Also there is no real justification in the introduction. Sex differences are important and need to be highlighted in the beginning of the manuscript.

Minor Comments

How was stress determined in the mice? Was corticosterone measured?

Usually western-pattern diet is referred to as a Western diet-or if this is different how is it different?

Author Response

Point 1: How was stress determined in the mice? Was corticosterone measured?

Response 1: Corticosterone was not measured. We put mice through well validates stressors and observed behavior. However, assessment of corticosterone would have been helpful for verification.

Point 2: Usually western-pattern diet is referred to as a Western diet-or if this is different how is it different?

Response 2: This is the same thing. It is the diet provided by Research Labs.

Reviewer 4 Report

In the reviewed manuscript entitled „The Modification of Offspring Stress-Related Behavior and Gene Expression by Diet”, the authors assess how a western diet interacts with prenatal stress to impact the expression of anxiety-like behavior and expression of the glucocorticoid and dopamine receptors in the offspring brain. This is an important issue due to the increasing number of people with the problem of mental disorders including anxiety. Before being accepted for publication, it is worth considering some additions and correcting editorial mistakes.

1. I suggest clarifying the title of the publication, taking into account the type of diet studied, and indicating the genes on which the authors focused, which will make it easier for interested readers to understand the assumptions of the work.

2. In the Results section or in the supplement, it would be useful to include data on the body weight of dams and offspring in the study groups and data on food intake, which would give a broader picture of the model used.

3. In the version received for review, the figures are not marked with statistical significance (*, **, ***), which are indicated in the description of the figures. It is also worth correcting the size of individual graphs in a given figure so that they are the same (e.g., Figure 2 A and B; Figure 4 C and D). In the description of the figures and in the methodology, please indicate the number of animals used for each behavioral and molecular experiment.

4. In the second paragraph of the discussion, it should be specified in which brain structure the changes in gene expression were observed (page 11, line 363).

5. In the discussion, it is worth considering supplementing the literature with recent studies on the influence of a maternal high-fat / high-sugar diet during pregnancy and lactation on anxiety-like behavior in the offspring and the influence on the predisposition to develop an addiction to psychostimulants mentioned by the authors (e.g., https://doi.org/10.1096/fj.202000163R; doi: 10.1007/s00213-007-1008-4; https://doi.org/10.1016/j.dcn.2020.100879; 10.1016/j.psyneuen.2017.05.003)

6. The discussion lacks an attempt to explain the reasons for the differences in the observed results for male and female offspring. It is also worth indicating more clearly why the research on the expression of selected genes was limited to one brain structure.

7. In the Materials and Methods section, reference is made to Figure 1, which provides a schematic of the experiment, which would be helpful to the readers. However, there is no such scheme in the reviewed version of the work.

8. The supplement should include the composition of the diets used in the study, which will be important for the readers, especially because the studied diets originated from different producers, which could have influenced the differences, for example, in the composition of micronutrients.

9. Please add a comment, on why PNDs 80-83 were selected for behavioral and molecular studies, and whether brains were collected from animals that were behaviorally tested or from another cohort.

10. Please check the manuscript carefully for any editorial errors that appear quite frequently in the text:

- please check the correctness of the use of the signs ">" in section 2.2.3 in the description of the results of the statistical analyzes

- consistently use abbreviations, if they were introduced at the beginning of the text, e.g., DA-ergic or Nacc

- missing ')' at the end of the sentence (page 5, line 190)

- no space at the end of the sentence (page 6, line 214)

- standardize in the manuscript: 3-way or three-way ANOVA

- please remove 'Add (A), (B), (C) info as for other captions.]' from the caption of Figure 8

- please italicized the name of genes in the text

Author Response

Point 1: I suggest clarifying the title of the publication, taking into account the type of diet studied, and indicating the genes on which the authors focused, which will make it easier for interested readers to understand the assumptions of the work.

Response 1: Thank you for this helpful suggestion. I have updated the title.

Point 2: In the Results section or in the supplement, it would be useful to include data on the body weight of dams and offspring in the study groups and data on food intake, which would give a broader picture of the model used.

Response 2: I have included data on body weights of the offspring and dams. However, we did not measure food intake.

Point 3: In the version received for review, the figures are not marked with statistical significance (*, **, ***), which are indicated in the description of the figures. It is also worth correcting the size of individual graphs in a given figure so that they are the same (e.g., Figure 2 A and B; Figure 4 C and D). In the description of the figures and in the methodology, please indicate the number of animals used for each behavioral and molecular experiment.

Response 3: There was apparently an issue with the format of my figures. I have updated the file types, so they should appear correctly now. I have also added n’s to the figures.

Point 4: In the second paragraph of the discussion, it should be specified in which brain structure the changes in gene expression were observed (page 11, line 363).

Response 4: Thank you for bringing this to our attention. This has been corrected.

Point 5: In the discussion, it is worth considering supplementing the literature with recent studies on the influence of a maternal high-fat / high-sugar diet during pregnancy and lactation on anxiety-like behavior in the offspring and the influence on the predisposition to develop an addiction to psychostimulants mentioned by the authors (e.g., https://doi.org/10.1096/fj.202000163R; doi: 10.1007/s00213-007-1008-4; https://doi.org/10.1016/j.dcn.2020.100879; 10.1016/j.psyneuen.2017.05.003)

Response 5: Thank you for this helpful suggestion. References have been added.

Point 6: The discussion lacks an attempt to explain the reasons for the differences in the observed results for male and female offspring. It is also worth indicating more clearly why the research on the expression of selected genes was limited to one brain structure.

Response 6: Thank you for pointing out this oversight. Discussion regarding reason for sex differences has been added. We also provided reasoning for why the selected genes were limited to the brain regions they were.

Point 7: In the Materials and Methods section, reference is made to Figure 1, which provides a schematic of the experiment, which would be helpful to the readers. However, there is no such scheme in the reviewed version of the work.

Response 7: A schematic has now been included.

Point 8: The supplement should include the composition of the diets used in the study, which will be important for the readers, especially because the studied diets originated from different producers, which could have influenced the differences, for example, in the composition of micronutrients.

Response 8: I have included a figure with diet compositions. More detailed micronutrient information can be found at the cited websites for each respective diet.

Point 9: Please add a comment, on why PNDs 80-83 were selected for behavioral and molecular studies, and whether brains were collected from animals that were behaviorally tested or from another cohort.

Response 9: This information has been added.

Point 10: Please check the manuscript carefully for any editorial errors that appear quite frequently in the text:

10.1: please check the correctness of the use of the signs ">" in section 2.2.3 in the description of the results of the statistical analyzes

Response 10.1: This has been corrected

10.2: consistently use abbreviations, if they were introduced at the beginning of the text, e.g., DA-ergic or Nacc

Response 10.2: This has been corrected

10.3: missing ')' at the end of the sentence (page 5, line 190)

Response 10.3: This has been corrected

10.4: no space at the end of the sentence (page 6, line 214)

Response 10.4: This has been corrected

10.5: standardize in the manuscript: 3-way or three-way ANOVA

Response 10.5: This has been corrected.

10.6: please remove 'Add (A), (B), (C) info as for other captions.]' from the caption of Figure 8

Response 10.6: Thank you for catching this! It has been removed.

10.7: please italicized the name of genes in the text

Response 10.7: We believe we have now corrected all incidences of non-italicized genes in text.

Round 2

Reviewer 2 Report

Dear Authors

I consider that the manuscript has been significantly improved after the work of synthesis and organisation of the results, new figures and a change in the order of the different sections.

The title does accurately reflect the content of the article.

The experimental design and sampling are well explained and reproducible.

The language used is clear and fits the scientific style.

The bibliographic references are adequate and are limited to the last 10 years

Congratulations

Author Response

Thank you for taking the time to help us improve this paper. 

Reviewer 4 Report

The authors have adequately addressed my initial concerns about the manuscript. Thank you for your work in clarifying these points. Minor comments to the revision version of the manuscript ‘The Modification of Offspring Stress-Related Behavior and the Expression of Drd1, Drd2, and Nr3c1 by a Western-Pattern Diet in Mus Musculus’, which should be included before publication:

1. The symbols for p < 0.01, p < 0.001 and p < 0.0001 should be unified in all figures and their descriptions (for example in Figure 4 ** p < 0.001, while in Figure 5 *** p < 0.001 or in Figures 4 and 7 * p < 0.05 and in Figure 5 * p < 0.01. In addition, the description of Figure 1 should be supplemented with an explanation of symbols ****.

2. In the description of the results, pay attention to the signs > and <. The authors mentioning significant statistical changes use the sign p > instead of p < (pages 5, 12, 16 - description of Figure 9).

Author Response

  1. The symbols for p < 0.01, p < 0.001 and p < 0.0001 should be unified in all figures and their descriptions (for example in Figure 4 ** p < 0.001, while in Figure 5 *** p < 0.001 or in Figures 4 and 7 * p < 0.05 and in Figure 5 * p < 0.01. In addition, the description of Figure 1 should be supplemented with an explanation of symbols ****.

Thanks for pointing this out. We have corrected this. 

  1. In the description of the results, pay attention to the signs > and <. The authors mentioning significant statistical changes use the sign p > instead of p < (pages 5, 12, 16 - description of Figure 9).

We apologize for missing these, as you did point this out in the last round. We have corrected the remainder of these mistakes.